# Preparation of BiOCl/Bi_2_WO_6_ Photocatalyst for Efficient Fixation on Cotton Fabric: Applications in UV Shielding and Self-Cleaning Performances

**DOI:** 10.3390/ma14227002

**Published:** 2021-11-18

**Authors:** Jiayi Chen, Kuang Wang, Jialong Tian, Wenhui Yu, Yujie Chen, Na Li, Zhenming Qi, Chunxia Wang

**Affiliations:** 1College of Textile and Clothing, Yancheng Institute of Technology, Yancheng 224051, China; jychen1982@163.com (J.C.); wkmineking@163.com (K.W.); jialongtian1996@163.com (J.T.); Eleanor980119@163.com (W.Y.); 2012310002@stmail.ntu.edu.cn (Y.C.); linahww@163.com (N.L.); stoner28@126.com (Z.Q.); 2School of Textile Science and Engineering, Jiangnan University, Wuxi 214122, China; 3School of Textile and Clothing, Nantong University, Nantong 226019, China

**Keywords:** BiOCl/Bi_2_WO_6_, cotton fabric, UV shielding, photocatalysis, self-cleaning

## Abstract

In this work, a visible-light-driven BiOCl/Bi_2_WO_6_ photocatalyst was obtained via a facile hydrothermal method and characterized by X-ray diffraction (XRD), scanning electron microscopy (SEM), energy-dispersive spectrometry (EDS), X-ray photoelectron spectroscopy (XPS), ultraviolet/visible light diffuse reflection spectroscopy (UV/Vis), and photocurrent (PC). BiOCl/Bi_2_WO_6_ was modified with (3-chloro-2-hydroxypropyl) trimethyl ammonium chloride to obtain the cationized BiOCl/Bi_2_WO_6_. Cotton fabric was pretreated with sodium hydroxide (NaOH) and sodium chloroacetate solution to obtain carboxymethylated cotton fabric, which was further reacted with cationized BiOCl/Bi_2_WO_6_ to achieve finished cotton fabric. The cotton fabrics were characterized by Fourier-transform infrared spectroscopy (FT-IR), XRD, SEM, and EDS. The photocatalytic activity of the BiOCl/Bi_2_WO_6_ photocatalyst and cotton fabrics was assessed by photocatalytic degradation of MB (methylene blue) solution under simulated visible light. The self-cleaning property of cotton fabrics was evaluated by removing MB solution and red-wine stains. Results revealed that the coated cotton fabrics exhibited appreciable photocatalytic and self-cleaning performance. In addition, anti-UV studies showed that the finished cotton fabrics had remarkable UV blocking properties in the UVA and UVB regions. Therefore, the finished cotton fabric with BiOCl/Bi_2_WO_6_ can provide a framework for the development of multifunctional textiles.

## 1. Introduction

Semiconductor-based photocatalysis is regarded as a promising and cost-effective approach to realize environmental decontamination [1,2,3,4,5]. Traditional semiconductor materials with a wide bandgap (e.g., TiO_2_, ZnO, and BiOCl) are widely utilized in practical application due to their commendable chemical stability, biosafety, photostability, and low cost [6,7,8,9]. BiOCl with a layered structure is a potential candidate for semiconductor photocatalysis owing to its superior photocatalytic activity. However, it displays a relatively low quantum efficiency in the visible wavelength range that hinders its practical implementation [8,9,10,11]. Another bismuth compound, namely, Bi_2_WO_6_, with its comparatively narrow bandgap, is capable of adsorbing visible light. Nevertheless, the rapid recombination of photogenerated electron–hole pairs in Bi_2_WO_6_ reduces its photocatalytic efficiency [12,13,14]. Interestingly, the lattice of Bi_2_WO_6_ and BiOCl can be well matched and form heterojunctions to promote the separation of photogenerated carriers [15,16]. Tahmasebi et al. fabricated a Bi_2_WO_6_/BiOCl composite through a facile hydrothermal method with the aid of hydrochloric acid (HCl) as a Cl source, and the obtained Bi_2_WO_6_/BiOCl exhibited preferable photocatalytic activity [17]. Visible light-responsive Bi_2_WO_6_/BiOCl heterojunctions were successfully prepared via a simple one-step hydrothermal method by Liang and his coworkers, and the results revealed that Bi_2_WO_6_/BiOCl had superior degradation efficiency under visible-light illumination than bare BiOCl and Bi_2_WO_6_ [18].

Cotton fabric is a very common natural fabric that has been widely used as the substrate of functional textiles due to its exceptional moisture absorption, breathability, comfortability, etc. [19,20,21,22]. Cotton fabrics have been finished with semiconductor materials by many researchers to obtain properties such as self-cleaning, UV-shielding, superhydrophobic, water–oil separating, and antibacterial [23,24,25]. Tudu et al. developed a superhydrophobic cotton fabric using a mixture of perfluorodecyltriethoxysilane (PFDTS) and TiO_2_ nanoparticles, and the coated cotton fabric presented outstanding self-cleaning, stain resistance, rust stain resistance, antiwater absorption, and antibacterial abilities [26]. BiOCl nanosheet-coated cotton fabric was obtained via a pad–dry–cure method by Jin and his colleagues; the finished cotton fabric showed appreciable UV protection and photocatalytic performance [27]. To the best of our knowledge, a cotton fabric finished with BiOCl/Bi_2_WO_6_ photocatalyst has not been reported to study its various functionalities, such as self-cleaning, UV-shielding, and photocatalytic properties.

In this study, we coupled BiOCl with Bi_2_WO_6_ through a facile one-step hydrothermal method and characterized it by XRD, SEM, EDS, XPS, UV/Vis DRS, PC, and photocatalytic activity. Thereafter, the cotton fabric was pretreated by NaOH and sodium chloroacetate solution to obtain carboxymethylated cotton fabric, and then finished with cationized BiOCl/Bi_2_WO_6_ to obtain multifunctional cotton fabric via double-dip/double-roll technology. The UV protection, photocatalytic, and self-cleaning properties of the cotton fabric coated with BiOCl/Bi_2_WO_6_ were investigated.

## 2. Experimental

### 2.1. Materials

Bismuth nitrate pentahydrate (Bi(NO_3_)_3_·5H_2_O, AR) and methylene blue (MB, AR) were purchased from Tianjin Damao Chemical Reagent Co., Ltd. (Tianjin, China). Sodium tungstate dehydrate (Na_2_WO_4_·2H_2_O, AR), acetic acid (CH_3_COOH, AR), and anhydrous ethanol (AR) were obtained from Sinopharm Chemical Reagent Co., Ltd. Sodium chloride (NaCl, AR) and sodium hydroxide (NaOH, AR) were provided by Jiangsu Tongsheng Chemical Reagent Co., Ltd. (Yixing, China). (3-Chloro-2-hydroxypropyl) trimethyl ammonium chloride (60%, AR) and sodium chloroacetate (AR) were purchased from Aladdin Biochemical Technology Co., Ltd. (Shanghai, China). Red wine (food grade) was bought from Greatwall Wine (Beijing, China). Cotton fabric (linear density, 20 tex × 20 tex; fabric density, 300 fiber/10 cm × 300 fiber/10 cm; weight, (120 ± 10) g/m^2^; plain) was obtained from Nantong Weiteng Textile Co., Ltd. (Nantong, China).

### 2.2. Preparation of BiOCl/Bi_2_WO_6_

In a typical preparation run [28], 5 mmol of Bi(NO_3_)_3_·5H_2_O was added to 40 mL of deionized water under ultrasonic oscillation for 30 min. An appropriate amount of NaCl was added to the aforementioned suspension and stirred for 30 min. Simultaneously, a stoichiometric amount of Na_2_WO_4_·2H_2_O was dissolved in 20 mL of deionized water to obtain a uniformly transparent solution and was added dropwise to the above mixture under constant magnetic stirring (Shanghai Sile Instruments Co. Ltd., Shanghai, China) for 60 min. The mixture was transferred to a 100 mL Teflon-lined stainless-steel autoclave and maintained at 160 °C for 8 h. After the reaction, the precipitate was centrifuged and rinsed with deionized water and anhydrous ethanol three times. Finally, the samples were dried at 60 °C for 10 h to acquire BiOCl/Bi_2_WO_6._ The prepared samples with Cl and W at molar ratios of 1:3, 1:2, 1:1, 2:1, and 3:1 were denoted Cl/W-1-3, Cl/W-1-2, Cl/W-1-1, Cl/W-2-1, and Cl/W-3-1, respectively. For comparison, bare Bi_2_WO_6_ and BiOCl were prepared without adding NaCl and Na_2_WO_4_·2H_2_O under the same conditions.

### 2.3. Modification of BiOCl/Bi_2_WO_6_

In brief, 5 g of (3-chloro-2-hydroxypropyl) trimethyl ammonium chloride was dissolved in 100 mL of deionized water under sonication for 30 min. The solution was added with 1 g of BiOCl/Bi_2_WO_6_ (Cl/W-1-2) and stirred for 30 min. The precipitate was collected by centrifugation (Changsha Xiangyi Centrifuge Instrument Co. Ltd., Changsha, China) and dried in an oven (Shanghai Jing Hong Laboratory instruments Co. Ltd., shanghai, China) at 60 °C for 12 h to obtain modified BiOCl/Bi_2_WO_6_ powder [29].

### 2.4. Pretreatment of Cotton Fabrics

Alkalized cotton fabrics: Cotton fabrics were cut into 6 cm × 6 cm pieces and washed in an ultrasound water bath to remove impurities. The pieces were immersed in NaOH aqueous solution (15 wt.%) at 20 °C for 10 min, rinsed with deionized water, and dried at 60 °C to obtain alkalized cotton fabrics.

Carboxymethylated cotton fabrics: In brief, 10 g of sodium chloroacetate was dissolved in 100 mL of ethanol/water (*v*/*v* = 6/4) solution. The prepared alkalized cotton fabrics were immersed in the above solution at 20 °C for 10 min and then dried at 70 °C for 1 h. The cotton substrates were washed by deionized water, treated with 2 g/L acetic acid solution, washed with deionized water, and dried at 60 °C to obtain carboxymethylated cotton fabrics [30].

### 2.5. Finishing of Cotton Fabrics

First, 0.1 g, 0.2 g, and 0.3 g of modified BiOCl/Bi_2_WO_6_ were dispersed in 50 mL of deionized water under ultrasonication (Kunshan Ultrasonic Instrument Co. Ltd., Suzhou, China) for 30 min to form a uniform suspension. From the schematic diagram of the preparation of BiOCl/Bi_2_WO_6_-cotton fabrics as shown in Figure 1, 1 g of the carboxymethylated cotton fabrics were soaked in the above suspension for 30 min. The cotton fabrics were finished via the double-dip/double-roll technology and dried at 60 °C to achieve cotton fabrics functionalized with BiOCl/Bi_2_WO_6_. The cotton fabrics were labeled Cl/W-0.1 cotton fabric, Cl/W-0.2 cotton fabric, and Cl/W-0.3 cotton fabric on the basis of the amount of modified BiOCl/Bi_2_WO_6_.

### 2.6. Characterization

X-ray diffraction was investigated using an X PERT3 POWDER diffractometer (PANalytical, Amsterdam, The Netherlands) to detect the crystalline structures of the as-prepared samples. Surface morphologies were characterized using a Nova Nano SEM 450 field-emission scanning electron microscope (FEI, Hillsboro, OR, USA). The SEM was equipped with a VANTAGE-DS1 energy-dispersive spectrometer (Oxford Instruments, Oxford, UK) to study the elemental compositions and their contents. X-ray photoelectron spectra were recorded on an ESCALAB 250Xi X-ray photoelectron spectrometer (Thermo Fisher Scientific, Waltham, MA, USA) to investigate the surface chemical states and their composite structures. UV/Vis diffuse reflectance spectra were inspected using TU-1901 UV/Vis diffuse reflectometer (Persee, Beijing, China) with the wavelength of 200–800 nm to record the optical absorption properties. Fourier-transform infrared spectra were recorded using a NEXUF-670 Fourier infrared spectrometer (NICOLET, Madison, GA, USA) to evaluate the chemical compositions. Photoelectric properties were measured on a CHI 660D electrochemical workstation (CHI, Shanghai, China) with a three-electrode system.

### 2.7. Photocatalytic Activity Measurement

Photocatalytic activities of photocatalyst and cotton fabric were determined via the photodegradation of aqueous MB and RhB (Rhodamine B) under simulated visible-light illumination. Typically, 0.01 g of the as-obtained photocatalyst was dispersed in 100 mL of 10 mg/L aqueous MB to form a suspension. Cotton samples (6 cm × 6 cm) were cut into 1 cm × 1 cm pieces and placed in 50 mL of 10 mg/L RhB solution under magnetic stirring (Shanghai Sile Instruments Co. Ltd., Shanghai, China). After being agitated in the dark for 30 min to reach the adsorption–desorption equilibrium, the suspension was irradiated and constantly stirred under a 300 W xenon lamp (Perfect Light, Beijing, China) with a 400 nm cut-off filter. About 3 mL of the reaction solution was taken at the 10 min time interval and then centrifuged for 5 min to remove photocatalyst particles. The absorbances of aqueous MB and RhB were investigated by UV/Vis spectrophotometry at 664 nm and 554 nm with deionized water as the reference at time intervals of 10 min during photocatalytic reaction. The photodegradation rate was calculated as the ratio of the concentration difference before and after irradiation compared with the concentration before irradiation (Equation (1)).
D = (1 − C_t_/C) × 100%,(1)
where D is the photodegradation rate, C is the absorbance of 10 mg/L aqueous MB and RhB, and C_t_ is the absorbance of aqueous MB and RhB after t min of illumination [31,32].

### 2.8. Assessment of Cotton Fabrics

UV-blocking performance of cotton fabrics was recorded by YG (B) 912E textile anti-UV performance tester (Darong, Wenzhou, China).

Self-cleaning performance was assessed with aqueous MB and red wine as the stains. In brief, 0.2 mL of 20 mg/L MB and red wine were dropped on the cotton fabrics (6 cm × 6 cm). After drying in air under ambient conditions, the samples were irradiated by simulated visible light [26]. The photos were taken at intervals of 40 min.

## 3. Results and Discussion

### 3.1. XRD Analysis of the Photocatalysts

XRD analysis is performed to investigate the crystal structure of the sample, and the powder XRD pattern is shown in Figure 2. The peaks at 11.98°, 24.09°, 25.86°, 32.49°, and 33.45° were assigned to the (001), (002), (101), (110), and (102) planes of tetragonal phase BiOCl (JCPDS No. 06-0249) [33,34]. The sharp and narrow peaks indicate the distinctive monophase of the good crystallinity of BiOCl. The diffraction peaks at 2θ = 28.31°, 32.79°, 47.16°, and 55.83° were attributed to the (113), (200), (220), and (313) planes of orthorhombic russelite phase Bi_2_WO_6_ (JCPDS No. 73-1126), and no other impurity peaks were identified [35]. BiOCl/Bi_2_WO_6_ in Figure 2b–e exhibited a consistent position in the XRD patterns, indicating that all the samples were BiOCl/Bi_2_WO_6_. Moreover, no diffraction peaks of BiOCl were detected in Cl/W-1-3 from Figure 2a, which may be due to the insufficient content of BiOCl/Bi_2_WO_6_.

### 3.2. SEM and EDS Analyses of the Photocatalysts

Figure 3 displays the SEM images and EDS of Cl/W-1-2. As displayed in Figure 3a,b, nanoflower-shaped Cl/W-1-2, 2–3 μm in diameter, was composed of hierarchical nanosheets. EDS is illustrated in Figure 3c to further reveal the chemical components of Cl/W-1-2. The weight percentages of Bi, W, O, and Cl accounted for 72.78%, 6.93%, 10.17%, and 10.13%, respectively, which verified the successful fabrication of BiOCl/Bi_2_WO_6_.

### 3.3. XPS Analysis

Figure 4 presents the XPS spectra of Cl/W-1-2. The XPS survey spectrum in Figure 4a revealed that the Cl/W-1-2 composite contains Bi, O, Cl, and W, while C was the residue of the apparatus. The contents of Bi, O, Cl, and W accounted for about 59.75%, 15.29%, 4.22%, and 20.73%, respectively. The peaks at 159.5 eV and 164.7 eV in Figure 4b corresponded to Bi 4f_7/2_ and Bi 4f_5/2_, respectively, which confirmed the presence of Bi^3+^ ions [36]. The peaks at 530.0 eV and 531.8 eV in Figure 4c could be ascribed to the lattice oxygen of the BiOCl/Bi_2_WO_6_ composite and the hydroxyl groups of the absorbed H_2_O molecules, respectively [37]. The Cl 2*p* spectrum in Figure 4d was fitted by two peaks with the binding energy at 199.8 eV and 198.2 eV, corresponding to Cl 2*p*_1/2_ and Cl 2*p*_3/2_ peaks, respectively [38]. The two peaks at 36 eV and 38.1 eV in Figure 4e were in accordance with W 4*f*_7/2_ and W 4*f*_5/2_, respectively, corresponding to W^6+^ in BiOCl/Bi_2_WO_6_ [39]. The XPS analysis proved that BiOCl was successfully coupled with Bi_2_WO_6_, which was consistent with the XRD patterns.

### 3.4. UV/Vis DRS Analysis of the Photocatalysts

UV/Vis DRS of Bi_2_WO_6_, Cl/W-1-2, and BiOCl are given in Figure 5. The absorption bandgap energy of the as-prepared samples was estimated according to E_g_ = 1240/λ_g_, where E_g_ is the bandgap and λ_g_ is the absorption edge [40]. As shown in Figure 5, the absorption edges of Bi_2_WO_6_, Cl/W-1-2, and BiOCl at about 444 nm, 415 nm, and 375 nm were inspected. The bandgaps of Bi_2_WO_6_, Cl/W-1-2 and BiOCl were calculated to be 2.79 eV, 2.98 eV, and 3.31 eV, respectively. Compared with BiOCl, Cl/W-1-2 exhibited a relatively narrower bandgap and, therefore, an expanded photoreponse range.

### 3.5. PC Analysis of the Photocatalysts

The photoelectrochemical properties of the photocatalysts were investigated to determine the photocatalytic activity under simulated visible light. PC intensity is an index of recombination efficiency of photogenerated electron–hole pairs. A higher PC intensity denotes a lower recombination of photogenerated carriers. As shown in Figure 6, the PC intensity of Cl/W-1-2 was higher than that of Bi_2_WO_6_ and BiOCl. This might be explained by the increasing interfacial charge carrier transfer efficiency and the decreasing photogenerated carrier recombination, owing to the combination of Bi_2_WO_6_ and BiOCl. The result showed that Cl/W-1-2 was beneficial to the photocatalytic degradation of pollutants.

### 3.6. Photocatalytic Activity of the Photocatalysts

Figure 7 displays the photocatalytic activities and pseudo-first-order kinetic models of the as-prepared samples under simulated visible-light irradiation. As shown in Figure 7a, BiOCl showed poor photocatalytic efficiency, while the other photocatalysts exhibited definite adsorption of MB solution in the dark reaction. The photodegradation rate for BiOCl/Bi_2_WO_6_ increased with the increasing content of W. The photocatalyst had no obvious change when the molar ratio of Cl to W reached 1:2. Furthermore, Cl/W-3-1 showed a higher degradation rate than BiOCl but a lower one than Bi_2_WO_6_, which may be attributed to the insufficient content of W. The heterostructure of BiOCl/Bi_2_WO_6_ restrained the recombination of electron–hole pairs to increase the surface active sites, thereby enhancing the photocatalytic capacity. Cl/W-1-2 had good photodegradation of MB as high as 90% in 60 min. Hence, Cl/W-1-2 ranked the first in the photodegradation of MB under visible-light irradiation.

### 3.7. XRD Analysis of Cotton Fabrics

Figure 8 presents the XRD patterns of cotton fabric, Cl/W-0.3 cotton fabric, and Cl/W-1-2. As shown in Figure 8a, the diffraction peaks at 2θ of 14.9°, 16.6°, 22.6°, and 34.4° were observed in cotton fabric, corresponding to the (101), (110), (002), and (040) planes of cellulose [41,42]. As shown in Figure 8c, the diffraction peaks at 2θ = 28.3°, 36.7°, 47.1°, and 53.5° were indexed to the (113), (016), (220), and (131) planes of Bi_2_WO_6_ (JCPDS No. 73-1126), while the diffraction peaks at 2θ of 11.9°, 25.8°, 32.5°, 58.6°, and 74.9° were related to the (001), (101), (110), (212), and (214) planes of BiOCl (JCPDS NO. 06-0249). In the XRD pattern of Cl/W-0.3 cotton fabric in Figure 8b, the (101), (110), (002), and (040) planes of the cellulose, as well as the (001), (101), (110), (212), and (220) planes of the modified BiOCl/Bi_2_WO_6_ photocatalyst, were observed. According to the above analysis, the modified BiOCl/Bi_2_WO_6_ was successfully loaded onto cotton fabric.

### 3.8. FT-IR Analysis

The typical FT-IR spectra of cotton fabric, Cl/W-1-2, and Cl/W-0.3 measured within 400–4000 cm^−1^ are illustrated in Figure 9. For the cotton fabric in Figure 9a, the absorption peaks at 3440 cm^−1^ and 2910 cm^−1^ were respectively derived from the stretching vibrations of –OH groups and –CH/CH_2_ groups in cotton fabric. The absorption peak at 1636 cm^−1^ was credited to the absorption peak of water molecules on the cotton surface. The absorption peaks at 1110 cm^−1^ and 1156 cm^−1^ were related to the symmetrical and asymmetrical stretching vibrations of C–O–C in cotton fabric. Cl/W-1-2 in Figure 9b also contained –OH groups, whose absorption peak of stretching vibration corresponded to 3440 cm^−1^. The peak at 1607 cm^−1^ was generated by the absorption peak of water molecules. The Bi–O absorption peak and W–O asymmetrical stretching vibration peak appeared at 400–636 cm^−1^. After loading with modified BiOCl/Bi_2_WO_6_, Cl/W-0.3 cotton fabric in Figure 9c presented a similar shape and position of absorption peaks as cotton fabric between 1093 and 4000 cm^−1^. Furthermore, the absorption bands at 400–639 cm^−1^ were assigned to Cl/W-1-2. The results showed that the BiOCl/Bi_2_WO_6_ photocatalyst was successfully loaded on cotton fabric [34].

### 3.9. SEM and EDS Analyses

The SEM images and EDS of cotton fabrics are presented in Figure 10. As shown in Figure 10a,c, the cotton fabric had a smooth surface before modification. After being treated with cationized BiOCl/Bi_2_WO_6_, Cl/W-0.3 cotton fabric showed a rough appearance due to the loading of numerous nanoparticles on its surface. As shown in Figure 10e, the EDS images confirmed the presence of Cl, W, Bi, C, and O on the Cl/W-0.3 cotton fabric. The weight fractions of Cl, W, and Bi were about 0.61%, 47.12% and 6.85%, respectively. Therefore, the SEM images and EDS confirmed the loading of the BiOCl/Bi_2_WO_6_ photocatalyst on cotton fabric.

### 3.10. Ultraviolet Resistance Evaluation

The ultraviolet resistance property of all the fabrics is recorded in Table 1. The cotton fabric had higher UVA and UVB levels than the finished cotton fabrics. The UPF (ultraviolet protection factor) of cotton fabric was only 5.95, which demonstrated poor ultraviolet resistance compared with the finished cotton fabrics. After modification with BiOCl/Bi_2_WO_6_ photocatalyst, the UVA and UVB of cotton fabrics obviously decreased, while the UPF significantly increased. In particular, Cl/W-0.3 cotton fabric exhibited the optimal ultraviolet resistance performance, with UPF reaching 40.15.

### 3.11. Photocatalytic Activity of the Cotton Fabrics

As shown in Figure 11a, with the extension of irradiation time, the photodegradation efficiency of cotton fabric to RhB was basically unchanged, while Cl/W-0.3 cotton fabric had a high photodegradation rate of 97.06%. Therefore, the finished cotton fabric had remarkable photocatalytic performance. As shown in Figure 11b, the pseudo-first-order kinetic constant of the Cl/W-0.3 cotton fabric was about 368.9 × 10^−4^ min^−1^, which was about 51.76 times that of cotton fabric (7.12702 × 10^−4^ min^−1^). Hence, the finishing of BiOCl/Bi_2_WO_6_ onto the cotton fabric increased the photocatalytic rate.

### 3.12. Self-Cleaning Evaluation

Figure 12 shows the self-cleaning effects of cotton fabric and Cl/W-0.3 cotton fabric to organic pollutants under simulated visible-light irradiation. Cotton fabric faded indistinctively after irradiation for 120 min. For Cl/W-0.3 cotton fabric, after 40 min of exposure to visible light, the red wine and MB solution showed slight decoloration. After 120 min of irradiation, the solutions stained with red wine and MB were both obviously discolored, indicating that the finished cotton fabric with BiOCl/Bi_2_WO_6_ had far better self-cleaning performance than cotton fabric [43].

## 4. Conclusions

A nanoflower-shaped BiOCl/Bi_2_WO_6_ photocatalyst was synthesized through a simple and efficient hydrothermal method. When the molar ratio of Cl to W was 1:2, the prepared BiOCl/Bi_2_WO_6_ exhibited higher photocatalytic activities than the bare BiOCl, Bi_2_WO_6_, and other BiOCl/Bi_2_WO_6_ composites under simulated visible-light illumination. Cotton fabric was carboxymethylated with NaOH and sodium chloroacetate solution and further finished with cationized BiOCl/Bi_2_WO_6_ to achieve the finished cotton fabric. The experimental results showed that the cotton fabric finished with the BiOCl/Bi_2_WO_6_ photocatalyst exhibited commendable UV shielding, photocatalytic, and self-cleaning properties. Hence, the BiOCl/Bi_2_WO_6_-finished cotton fabrics have broad application prospects in the textile, environmental purification, and medical industries.

## Figures and Tables

**Figure 1 materials-14-07002-f001:**
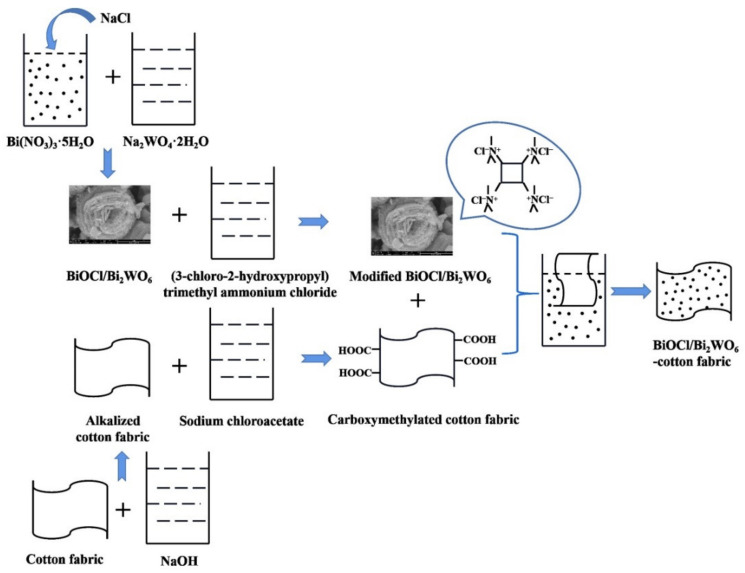
Schematic diagram of the preparation of BiOCl/Bi_2_WO_6_-cotton fabrics.

**Figure 2 materials-14-07002-f002:**
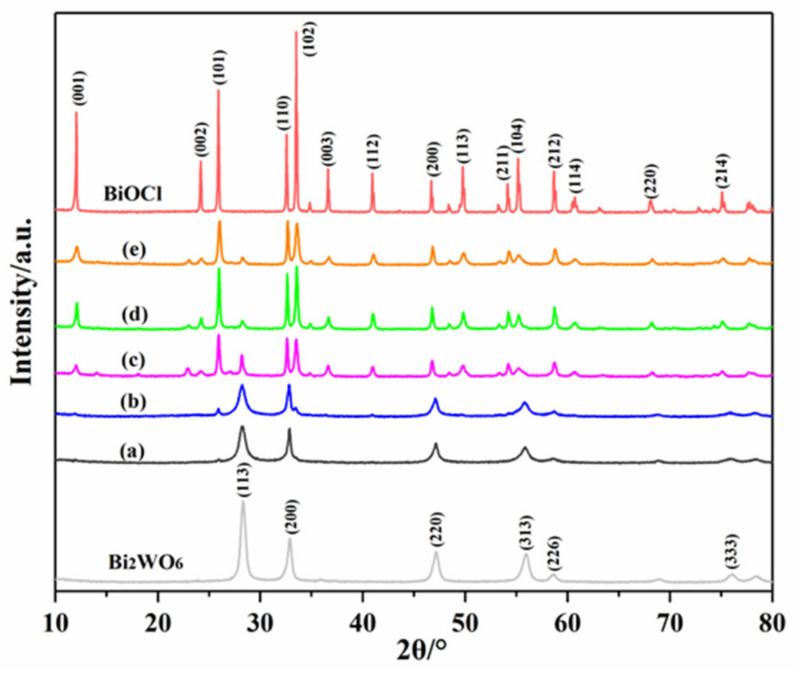
XRD patterns: Bi_2_WO_6_, (**a**) Cl/W-1-3, (**b**) Cl/W-1-2, (**c**) Cl/W-1-1, (**d**) Cl/W-2-1, (**e**) Cl/W-3-1, and BiOCl.

**Figure 3 materials-14-07002-f003:**
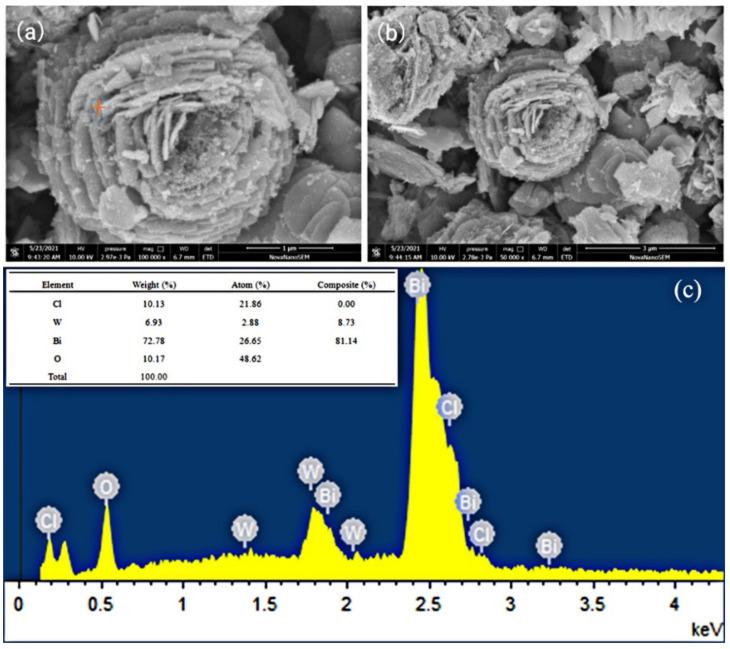
SEM imagesand EDS of Cl/W-1-2: (**a**) Cl/W-1-2 (×100,000), (**b**) Cl/W-1-2 (×50,000), and (**c**) EDS of Cl/W-1-2.

**Figure 4 materials-14-07002-f004:**
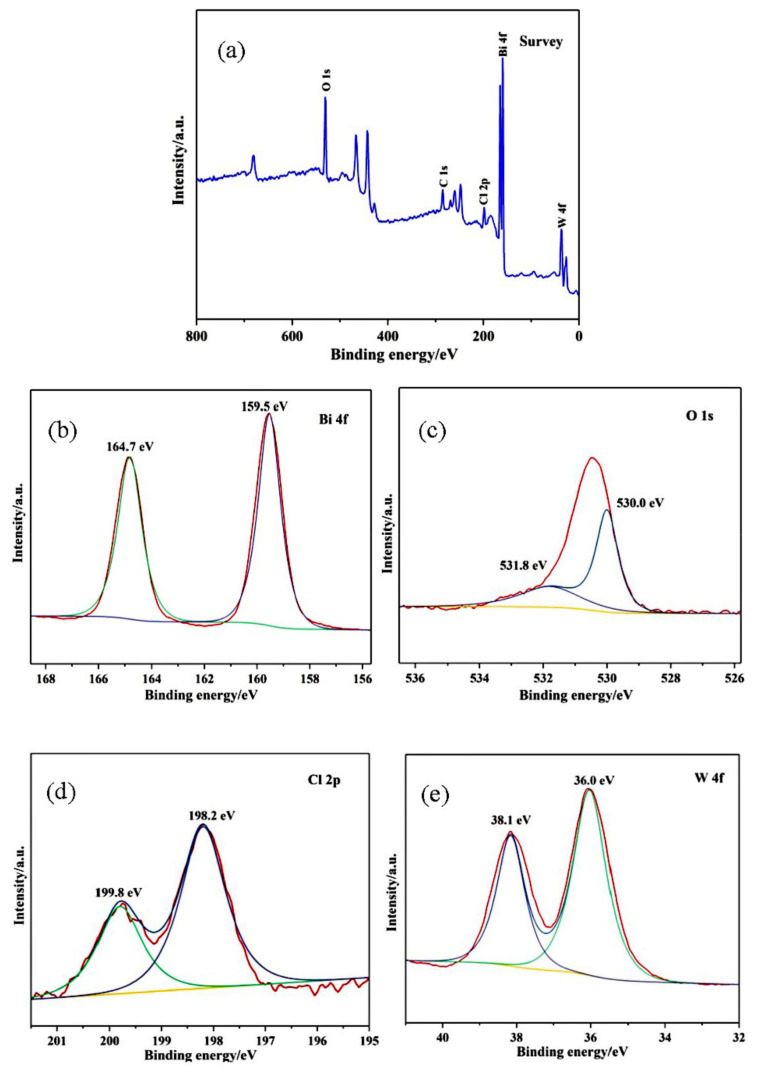
XPS spectra of Cl/W-1-2: (**a**) survey, (**b**) Bi 4*f*, (**c**) O 1*s*, (**d**) Cl 2*p*, and (**e**)W 4*f*.

**Figure 5 materials-14-07002-f005:**
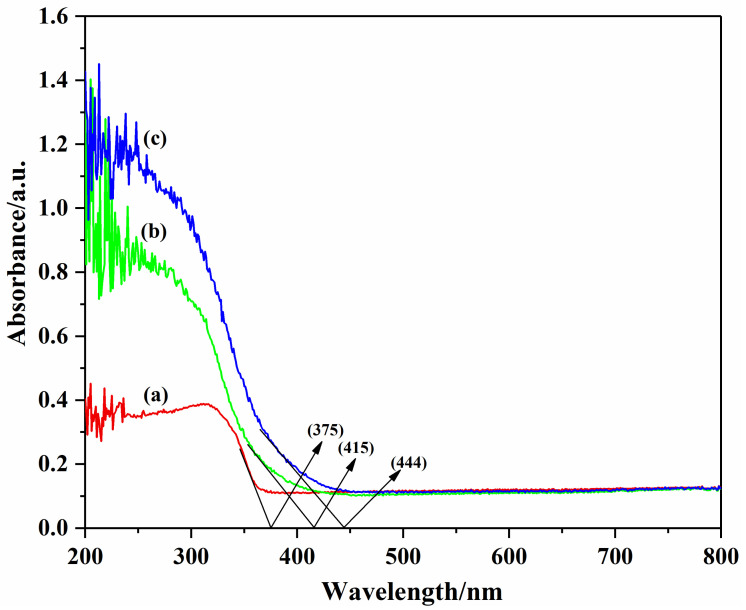
UV/Vis DRS of the photocatalysts: (**a**) BiOCl, (**b**) Cl/W-1-2, and (**c**) Bi_2_WO_6_.

**Figure 6 materials-14-07002-f006:**
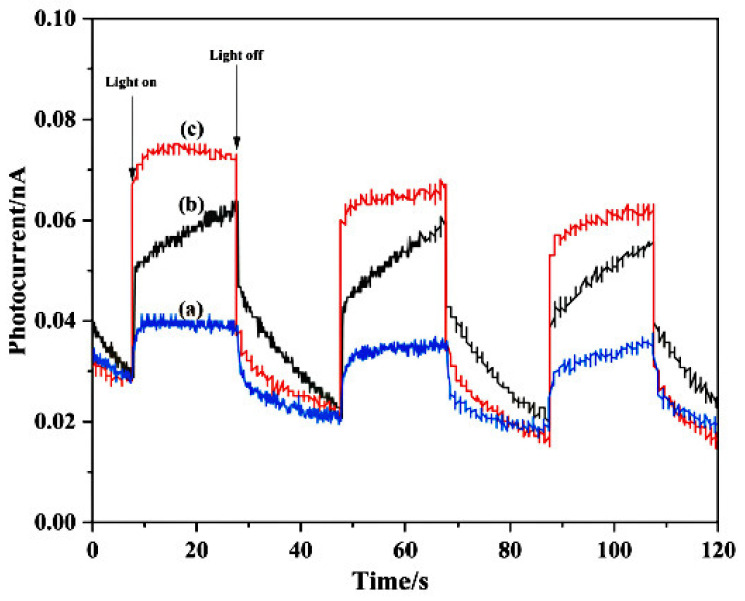
Photocurrent of the photocatalysts: (**a**) BiOCl, (**b**) Bi_2_WO_6_, and (**c**) Cl/W-1-2.

**Figure 7 materials-14-07002-f007:**
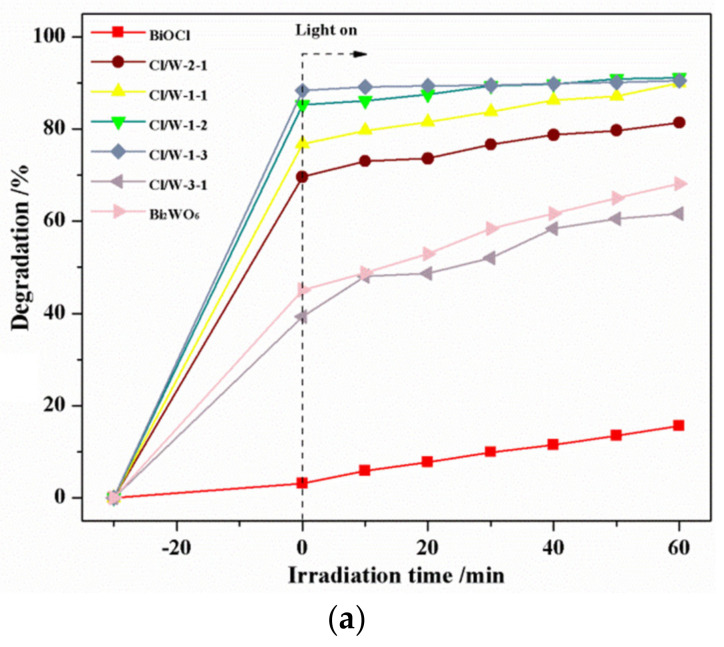
Photocatalytic degradation for MB by photocatalysts: (**a**) photodegradation curves and (**b**) pseudo-first-order kinetic curves.

**Figure 8 materials-14-07002-f008:**
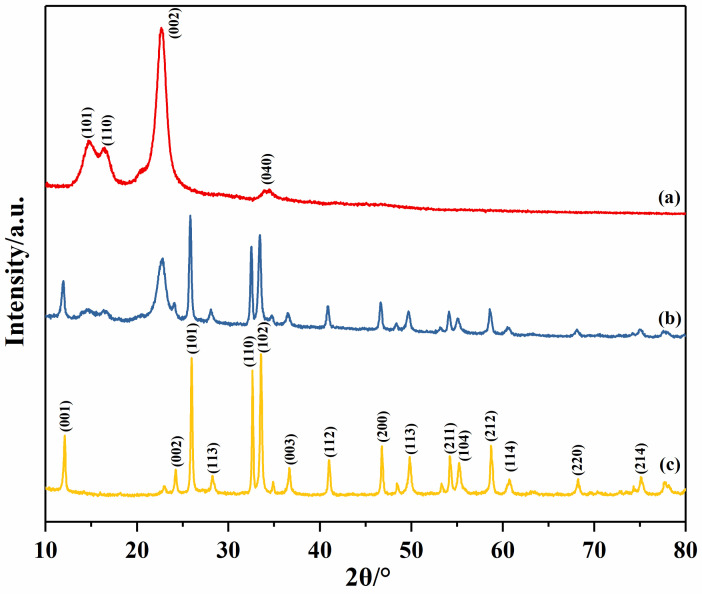
XRD patterns: (**a**) cotton fabric, (**b**) Cl/W-0.3 cotton fabric, and (**c**) Cl/W-1-2.

**Figure 9 materials-14-07002-f009:**
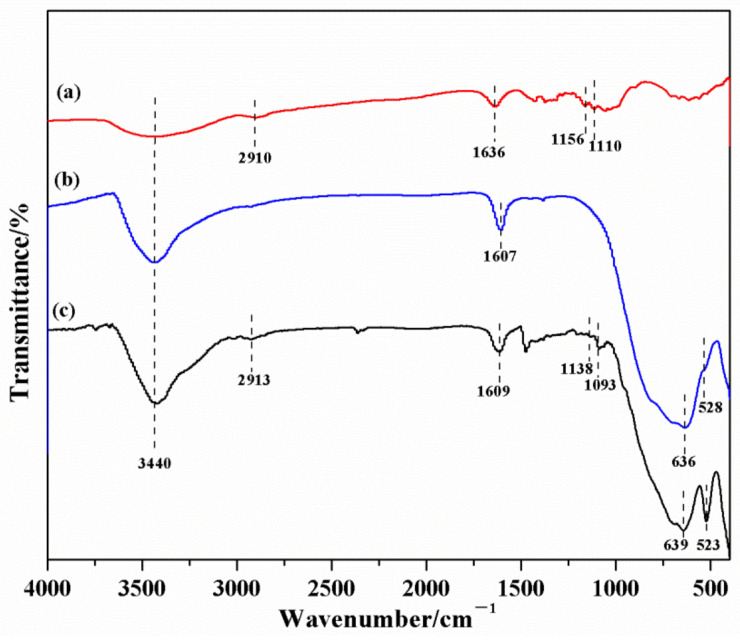
FTIR spectra: (**a**) cotton fabric, (**b**) Cl/W-1-2, and (**c**) Cl/W-0.3 cotton fabric.

**Figure 10 materials-14-07002-f010:**
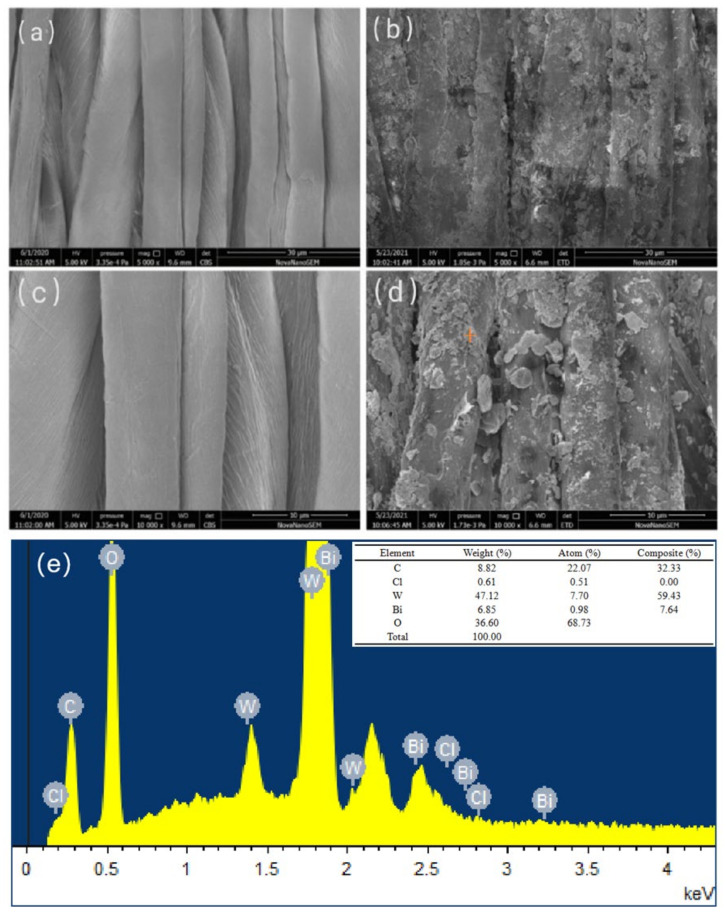
SEM images and EDS of cotton fabrics: (**a**) cotton fabric (×5000), (**b**) Cl/W-0.3 cotton fabric (×5000), (**c**) cotton fabric (×10,000), (**d**) Cl/W-0.3 cotton fabric (×10,000), and (**e**) EDS of Cl/W-0.3 cotton fabric.

**Figure 11 materials-14-07002-f011:**
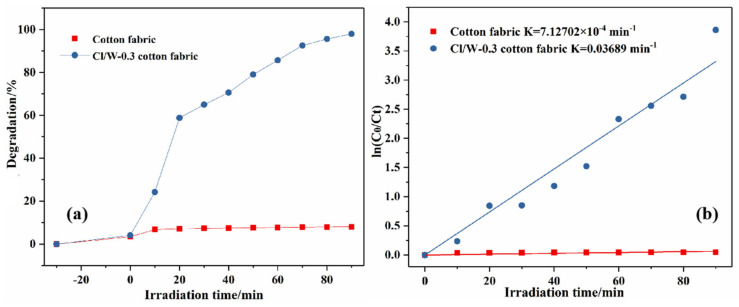
Photocatalytic degradation for RhB by cotton fabrics: (**a**) photodegradation curves and (**b**) pseudo-first-order kinetic curves.

**Figure 12 materials-14-07002-f012:**
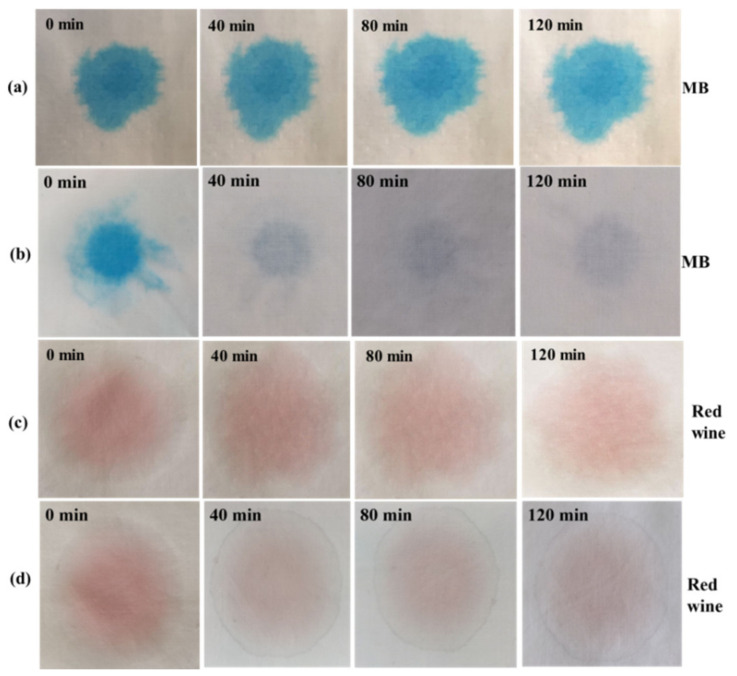
Self-cleaning effects of (**a**,**c**) cotton fabric, and (**b**,**d**) Cl/W-0.3 cotton fabric.

**Table 1 materials-14-07002-t001:** Ultraviolet resistance evaluation.

Sample	UVA (%)	UVB (%)	UPF
Cotton fabric	20	15.45	5.95
Cl/W-0.1 cotton fabric	8.36	2.63	31.92
Cl/W-0.2 cotton fabric	5.77	3.10	37.96
Cl/W-0.3 cotton fabric	3.51	3.45	40.15

## Data Availability

Not applicable.

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
