# Peer review of "Preparation of BiOCl/Bi2WO6 Photocatalyst for Efficient Fixation on Cotton Fabric: Applications in UV Shielding and Self-Cleaning Performances"

_materials, 2021, doi:10.3390/ma14227002_

Round 1
Reviewer 1 Report
The paper is devoted to the preparation and testing of the photocatalyst based on the bismuth compounds. The paper might be interesting to the scientists, working in the environmental control and purification area, medicine, and others. The paper may be published after some corrections:
1) Self-cleaning effects of Cl/W-0.3 cotton fabric should be better demonstrated compared to the normal cotton fabric.
2) How long is the shelf life of the prepared fabric? What are the storage conditions for the prepared material?
3) Is the catalyst regeneration process is necessary and possible? How long may be the fabric utilized? What is its efficiency?
4) Did the authors detected the degradation products of the colorants?
Also, some comments concerning methodological questions and language corrections are added in the article pdf file.

Author Response
Thank you for taking your time. Please see the attachment.

Reviewer 2 Report
The paper by Jiayi Chen et al. reports the hydrothermal preparation of BiOCl/Bi2WO6 photocatalysts and their fixation on cotton fabrics to obtain their functionality of UV shielding and self-cleaning. This research is very interesting and actual and could have a significant impact on the textile and clothing industry. The results were presented in a logical and very readable manner, for which the authors deserve praise. The minor improvements of the manuscript are needed:
1. Please check the writing errors, like font change in the 132 line, DES misspell in line 180, respectively in line 147, font miswrites in line 132, etc.
2. Add some more references in the Introduction section about the examples of heterostructured semiconductor materials used in photocatalysis, like Scientific Reports 10 (2020) 11903, Ceram. Int. 42 (2016) 3575-3583, Chinese J. Catal. 40 (2019) 796, J. Hazard. Mater. 390 (2020) 121623, Catalysts 11 (2021) 1054, Scientific Reports 10 (2020) 18401.
3. Add a schematic diagram of the preparation of BiOCl/Bi2WO6 cotton fabric.
4. Mark exactly the point (area) on the SEM images (Figure 2 a-b) from which the presented EDS analysis (Figure 2c) originates.
5. Is the XRD pattern of Cl/W-1-2 in Figure 1b the same as the XRD pattern of Cl/W-1-2 shown in Figure 7c?
6. Please provide, if available, the FT-IR spectrum of Cl/W-1-2 alongside the ones for cotton fabric and Cl/W-0.3 in Figure 8.
7. Please, mark exactly the point (area) on the SEM images (Figure 9 b,d) from which the presented EDS analysis (Figure 9e) originates.
8. Please describe the UPF abbreviation when first mentioned and then use it further in the text. Check the whole text for similar.
9. Please mention in the materials characterization sections that you chose to show only the characterization results of samples Cl/W-1-2 and Cl/W-0.3 according to the photocatalytic and resistance evaluation results and their best performance among other tested samples because the readers could be confused why only those results are shown, before coming to the sections where that becomes obvious i.e. the order of the presented results requires explanation.
Author Response

(The authors gave the same response as above.)

Reviewer 3 Report
The manuscript entitled „Preparation of BiOCl/Bi2WO6 photocatalyst for efficient fixation on cotton fabric: Applications in UV shielding and self-cleaning performances” describes the preparation and characterization of bismuth photocatalysts as well as their application on cotton fabrics and evaluation of the obtained materials. The topic is fairly relevant and it fits within the scope of the Journal. The introduction is comprehensive and covers the recent literature. The study is interesting and the paper has the correct form with logical sections; nevertheless, I would like to make some considerations about the work.
1. Although the language is mostly understandable, I suggest using more formal vocabulary (eg. line 24: I would change “adorable” to “remarkable” or “extraordinary”; line 247: “decorated onto” to “loaded on”, “covered” or “coated” and line 268: “decoration” to “loading” or “coverage” – similar in lines 111, 289: “finishing“, 299, 308-310, 312: “finished”)
2. Some phrases are not clear, eg. lines 275/276: “The cotton fabric had highest UVA and UVB than the finished cotton fabrics” 287-289: “As shown in Figure 10(b), the pseudo-first-order kinetic constant of the Cl/W-0.3 cotton fabric was about 368.9×10-4 min-1, which was about 51.76 times of the cotton fabric (7.12702×10-4 min-1)”
3. Line 276: The abbreviation “UPF” is not described in the manuscript.
4. Line 153: Was the red wine applied in solution or undiluted?
5. Have the authors performed any toxicity studies or are there any literature data available that are devoted to this issue?
6. The authors could discuss in detail the potential applications of the proposed materials with specific examples. Please, comment on how the fabrics can be used in practice.
To sum up, I recommend the paper for publication in Materials after revisions.
Author Response

(The authors gave the same response as above.)

Reviewer 4 Report
This is an excellent piece of work. Please answer to the following questios.
1 - It is said: "evaluated by removing MB solution and red wine stains". Please indicate the meaning fo "MB solution".
2 - Please add a sentence explaining what is the degradation rate in Figure 6
Author Response

(The authors gave the same response as above.)

Reviewer 5 Report
The presented manuscript includes the study of the preparation of BiOCl/Bi2WO6 photocatalyst for efficient fixation on cotton fabric: applications in UV shielding and self-cleaning performances.
The manuscript is well written and structured. The results of the work are presented on a good level, but, a couple of corrections could be suggested to prove the authors’ idea.
Q1. What was the reason to choose the dose of photocatalysts 0.1 g per 100 mL of solution (1 g/L)? The doses of 100-300 mg/L are the most common effective dose in the majority of the photocatalysts papers.
Q2. There is a lack of comparison of results with similar published ones. Also, please add recent papers of 2021 and maybe 2022 years.
Author Response

(The authors gave the same response as above.)
